# Distinct Characteristics and Clinical Outcomes to Predict the Emergence of *MET* Amplification in Patients with Non-Small Cell Lung Cancer Who Developed Resistance after Treatment with Epidermal Growth Factor Receptor Tyrosine Kinase Inhibitors

**DOI:** 10.3390/cancers13123096

**Published:** 2021-06-21

**Authors:** Beung-Chul Ahn, Ji Hyun Lee, Min Hwan Kim, Kyoung-Ho Pyo, Choong-kun Lee, Sun Min Lim, Hye Ryun Kim, Byoung Chul Cho, Min Hee Hong

**Affiliations:** 1Yonsei Cancer Center, Division of Medical Oncology, Department of Internal Medicine, Yonsei University College of Medicine, Seoul 03722, Korea; abcduke@yuhs.ac (B.-C.A.); jhlee0811@yuhs.ac (J.H.L.); GEMGOON3691@yuhs.ac (M.H.K.); PKHPSH@yuhs.ac (K.-H.P.); CKLEE512@yuhs.ac (C.-k.L.); LIMLOVE2008@yuhs.ac (S.M.L.); nobelg@yuhs.ac (H.R.K.); CBC1971@yuhs.ac (B.C.C.); 2Center for Lung Cancer, Research Institute and Hospital, National Cancer Center, Goyang 10408, Korea; 3Severance Biomedical Science Institute, Yonsei University College of Medicine, Seoul 03722, Korea

**Keywords:** non-small cell lung cancer, epidermal growth factor receptor, tyrosine kinase inhibitor, *MET* amplification

## Abstract

**Simple Summary:**

*MET* amplification is one of the resistance determinants after EGFR-TKI therapy in EGFR mutant NSCLC. In this study, we evaluated the emergence of MET amplification after EGFR-TKI treatment failure. The median progression-free survival associated with the most recent EGFR-TKI treatment was shorter in MET amplification-positive patients than in negative patients. Smoking history and less intracranial progression are associated with MET amplification. Suboptimal responses with previous EGFR-TKI are associated with MET amplification. Proper MET amplification screening for therapeutic targeting is needed.

**Abstract:**

Objectives: Patients with epidermal growth factor receptor (*EGFR*) mutant non-small cell lung cancer (NSCLC) ultimately acquire resistance to *EGFR* tyrosine kinase inhibitors (TKIs) during treatment. In 5–22% of these patients, resistance is mediated by aberrant mesenchymal epithelial transition factor (*MET*) gene amplification. Here, we evaluated the emergence of *MET* amplification after EGFR-TKI treatment failure based on clinical parameters. Materials and Methods: We retrospectively analyzed 186 patients with advanced *EGFR*-mutant NSCLC for *MET* amplification status by in situ hybridization (ISH) assay after EGFR-TKI failure. We collected information including baseline patient characteristics, metastatic locations and generation, line, and progression-free survival (PFS) of EGFR-TKI used before *MET* evaluation. Multivariate logistic regression analysis was conducted to evaluate associations between *MET* amplification status and clinical variables. Results: Regarding baseline *EGFR* mutations, exon 19 deletion was predominant (57.5%), followed by L858R mutation (37.1%). The proportions of *MET* ISH assays performed after first/second-generation and third-generation TKI failure were 66.7% and 33.1%, respectively. The median PFS for the most recent EGFR-TKI treatment was shorter in *MET* amplification-positive patients than in *MET* amplification-negative patients (median PFS 7.0 vs. 10.4 months, *p* = 0.004). Multivariate logistic regression demonstrated that a history of smoking, short PFS on the most recent TKI, and less intracranial progression were associated with a high probability of *MET* amplification (all *p* < 0.05). Conclusions: Our results demonstrated the distinct clinical characteristics of patients with *MET* amplification-positive NSCLC after EGFR-TKI therapy. Our clinical prediction can aid physicians in selecting patients eligible for *MET* amplification screening and therapeutic targeting.

## 1. Introduction

Patients with non-small cell lung cancer (NSCLC) with epidermal growth factor receptor (*EGFR*) gene mutations are defined as an important targetable molecular subset [1]. First- or second-generation EGFR tyrosine kinase inhibitor (EGFR-TKI) therapies in previous trials have been shown to induce robust tumor regression that lasts for 9–12 months on average [2,3]. In contrast, third-generation EGFR-TKIs, designed as antagonists to both T790M-mutated *EGFR* and conventional *EGFR* mutations (exon 19 deletion and L858R substitution), exhibit robust activity against T790M-mutant NSCLCs, showing progression after first-line EGFR-TKI therapy, with a progression-free survival (PFS) of 8.5 months and a prolonged PFS of 17.2 months in a upfront trial [4,5]. However, acquired resistance is inevitable in most patients during the course of EGFR-TKI therapy, and the molecular determinants of resistance are highly variable from patient to patient [6,7]. Moreover, recent studies elucidated uncommon *EGFR* mutation and coexisting genetic alteration might likely justify resistance to TKIs treatment. Different studies have assessed that the uncommon *EGFR* mutation and concurrent presence of mutation provide worse prognosis in *EGFR*-positive NSCLC patients treated with first-, second-, and third-generation TKIs [8,9].

Mesenchymal epithelial transition factor (MET) is a receptor for hepatocyte growth factor (HGF) in mammalian cells and transduces growth factor signals to downstream mitogen-activated protein kinase and phosphatidylinositol 3-kinase pathways through tyrosine kinase activity [10]. Aberrant MET activation promotes cellular proliferation, angiogenesis, and metastasis in many cancer types, including kidney, liver, stomach, breast, and brain cancers. Importantly, amplification of the *MET* gene has been reported to be involved in acquired resistance after EGFR-TKI therapy, accounting for 3–5% of resistance to first/second-generation EGFR-TKIs [11,12]. Resistance after first-line osimertinib therapy is even more enriched for *MET* amplification, which is the most common cause of resistance, observed in 15% of all cases [13]. Interestingly, the recent TATTON trial demonstrated the promising activity of the *MET* inhibitor salvolitinib in patients with *MET*-amplified NSCLC, establishing *MET* amplification as the most important actionable target after EGFR-TKI resistance besides T790M *EGFR* mutation [14,15]. However, the incidence of *MET* amplification is considerably low, limiting routine *MET* in situ hybridization (ISH) or mutation screening after EGFR-TKI failure.

NSCLC comprises heterogeneous molecular subtypes harboring mutations in various oncogenes, including *EGFR*, *ALK* receptor tyrosine kinase (*ALK*), *ROS* proto-oncogene 1 (*ROS1*), *RET*, B-Raf proto-oncogene, and human epidermal growth factor receptor 2. During analysis of the “long-tail” distribution patterns of mutations, previous studies have reported distinct clinicopathological characteristics of each genotype. For example, EGFR-mutant NSCLC is highly prevalent in never-smokers and Asian populations [16]. *ALK* rearrangement is associated with young patient age at diagnosis, signet ring cell pathology, and bronchioalveolar involvement pattern [17]. These features have facilitated the diagnosis of the different subtypes in clinical practice and enabled the focused molecular investigation of a wide range of clinical presentations of NSCLC. Moreover, recent studies have reported distinct clinical characteristics of patients with emergence of the T790M mutation during EGFR-TKI therapy [18]. However, the clinicopathological characteristics of *MET*-amplified NSCLC after EGFR-TKI failure have not been evaluated, and no predictive factors for *MET* amplification have been reported.

Accordingly, in this study, we retrospectively analyzed 186 consecutive patients with EGFR-mutant NSCLC who underwent *MET* ISH assays on rebiopsied tumors. We compared the clinicopathological parameters and therapy responses for the previous EGFR-TKI therapy between patients with *MET*-amplified and non-*MET*-amplified NSCLC. We ultimately aimed to develop a clinical prediction method for *MET* amplification after EGFR-TKI failure, which is an essential step for focused *MET* screening and subsequent therapeutic *MET* targeting in patients.

## 2. Methods

### 2.1. Patients

This study retrospectively analyzed 186 patients with advanced *EGFR*-mutant NSCLC who progressed on EGFR-TKI therapy and underwent *MET* ISH assays at Yonsei Cancer Center between April 2004 and March 2019. All patients received EGFR-TKI therapy, and post-treatment tumor biopsy was performed after documented progression. The clinicopathological data for the patients, including age, sex, smoking status, type of *EGFR* mutation, metastatic site, line of EGFR-TKI therapy, type of EGFR-TKIs used, and PFS on previous EGFR-TKI, were collected by electronic medical chart review. Informed consent for *MET* ISH analysis and use of data was obtained at the time of rebiopsy. This study was approved by the institutional review board of Severance Hospital (institutional review board approval No. 4-2019-0426).

### 2.2. Assessments

Chest computed tomography (CT) and abdominal CT were taken every 2–3 months during treatment. Besides regular assessment follow-up, additional imaging was performed according to the physician’s intention. The response to EGFR-TKI treatment was radiologically assessed according to the Response Evaluation Criteria in Solid Tumors, version 1.1. The objective response rate (ORR) was defined as the proportion of patients with complete response (CR) or partial response (PR), and the disease control rate (DCR) was defined as the proportion of patients with CR, PR, or stable disease. Progression-free survival (PFS) was defined as the time from the start of EGFR-TKI treatment to disease progression or death. Overall survival (OS) was defined as the start of TKI treatment initiation to the date of death. In order to characterize OS based on treatment lines and TKIs, we also measured OS from the date of *MET* ISH assays to the date of death.

### 2.3. EGFR Mutation and MET ISH Analyses

The diagnosis of NSCLC with non-squamous histology was confirmed by pathological and radiological examination. In all patients, the *EGFR* mutation status was evaluated using a PANA Mutyper detection kit. The *MET* amplification status was assessed using *MET* VENTANA dual probe sliver ISH assays, defining a *MET*/CEP7 signal ratio of greater than or equal to 2 or a *MET* average copy number in at least 50 enumerated cells of greater than or equal to 5 as *MET* amplification positive [19].

### 2.4. Statistical Analysis

The categorical parameters were compared between *MET* amplification-positive and -negative patients using chi-squared or Fisher’s exact tests as appropriate. Continuous variables were compared using Student’s *t*-tests. The PFS and OS were estimated using Kaplan–Meier curves, and subgroups were compared using log-rank tests. Logistic regression analyses were done to identify significant clinical factors predicting the emergence of *MET* amplification after EGFR-TKI therapy in the study cohort. Statistical analyses were performed with Statistical Package for Social Sciences (SPSS, Chicago, IL, USA) version 23.0 for Windows and GraphPad Prism version 8.0 (GraphPad Software, San Diego, CA, USA). Two-sided *p*-values of less than or equal to 0.05 were considered significant.

## 3. Results

### 3.1. Clinicopathological Characteristics of the Study Population

In total, 186 patients with advanced *EGFR*-mutant NSCLC were enrolled; the majority of patients were women (62.4%), and the median age was 61 years (range, 28–84 years). Most of the patients were never smokers (67.2%), and 0.0 pack-years of smoking was the median for the whole patient population. Regarding primary *EGFR* mutation status before *MET* ISH biopsy, the existence of exon 19 deletion mutation was predominant (57.5%), followed by L858R mutation (37.1%). Other mutations including Exon 20 insertion (p.A767_V769dup) (0.5%), G719X (1.1%), L861Q (2.2%), S768I (1.1%), and G179S/L861Q (0.5%) also existed. Almost all patients in the third-generation TKI cohort, except for six cases, had T790M mutation coexistence at the point of *MET* ISH assay owing to label use of third-generation TKIs in Korea. The six cases excluded from this regulation included one with off-label use of osimertinib, one with use of rociletinib during clinical trials (without T790M mutation), and three with use of mavelertinib during clinical trials (without T790M mutation). The proportions of *MET* ISH assays performed after first/second-generation (gefetinib, erlotinib, and afatinib) or third-generation (osimertinib, olmutinib, rociletinib, mavelertinib, and lazertinib) TKI failure were 66.7% and 33.3%, respectively. The proportions after failure of first-line and later-line EGFR-TKIs were 44.1% and 55.9%, respectively. At baseline, 78 (41.9%) patients had brain metastasis. At the time of *MET* ISH screening, 64 patients had progression of existing intracranial lesions, and 39 patients progressed with new lesions in the brain. Additionally, 8.1% (*n* = 15) of patients had liver metastasis at baseline. Eleven patients showed progression of existing lesions, and 28 patients had new liver metastasis at the time of rebiopsy.

*MET* amplification was identified in 30 (16.1%) of 186 patients. Patient characteristics based on *MET* amplification status are shown in Table 1, Appendix A. No significant differences in terms of patient characteristics, including sex, age, and liver metastasis status, were observed between the two groups. In contrast, smoking status (*p* = 0.013), previous TKI generation (*p* = 0.034), previous TKI line (*p* = 0.003), and intracranial progression at the time of TKI failure (*p* < 0.001) showed significant differences between the two groups. By the time of data lock (4 March 2021), 133 deaths (71.5%) had occurred, and 53 patients (28.5%) were alive. The median follow-up duration for all patients was 65.4 months.

### 3.2. Treatment Outcomes of EGFR-TKIs According to MET Status

The median OS (calculated from the date of *MET* ISH assays) for the total population was:

11.6 months (95% confidence interval [CI]: 9.7–13.5) (Figure 1A). The median OS from the initiation of EGFR-TKI therapy was 39.2 months (95% CI: 33.6–44.8). Only slight differences in the two OS attributes were observed after early initiation of EGFR-TKI treatment in most patients. There were no significant differences in OS (from the initiation of EGFR-TKI therapy) according to the *MET* amplification status (40.0 months [95% CI: 34.7–45.2] versus 34.7 months [95% CI: 21.5–47.88]; *p* = 0.53). However, it was difficult to interpret the actual prognostic effects of *MET* status because some patients (*n* = 23) were enrolled in *MET* inhibitor clinical trials.

The median PFS for the latest TKI in the total population was 10.2 months (95% CI: 9.3–11.1) (Figure 1B). Because of differences in PFS between first-/second- and third-generation TKIs, we analyzed these drugs separately. For patients receiving first-/second-generation TKIs, PFS was significantly shorter in the *MET*-positive group than in the negative group (7.0 versus 10.4 months; *p* = 0.003). However, in the third-generation TKI group, there were no significant differences in PFS (7.2 versus 11.0 months; *p* = 0.163), and only similar trends were observed (Figure 2).

### 3.3. Potential Predictors for MET Amplification

Table 2 shows the results of univariate and multivariate analysis by a logistic regression model to determine the predictive factors for *MET* amplification. Univariate analysis showed that smoking status, TKI generation, lines of EGFR-TKI therapy, PFS of the most recent TKI, and brain metastasis status at presentation were significant predictive factors. However, multivariate analysis considering all covariables identified only smoking status (hazard ratio [HR]: 3.475, 95% CI: 1.326–8.440; *p* = 0.011), PFS of most recent TKI (HR: 0.898, 95% CI: 0.833–0.967; *p* = 0.004), and intracranial progression status (HR: 0.138, 95% CI: 0.051–0.347; *p* < 0.001) as significant variables. The presence of smoking history, short PFS on latest TKI, and no intracranial progression at EGFR-TKI failure were associated with a high probability of *MET* amplification emergence.

The T790M mutation was not included in the logistic regression analysis. The reason is that in the case of acquiring the T790M mutation, since most of the patients were administered the 3rd generation TKI, multicollinearity occurred between the two variables.

### 3.4. Characteristics of Patients with MET Amplification

We further investigated the characteristics of patients with *MET* amplification (*n* = 30) based on the most recent TKI outcomes and progression site (Table 3). The ORR of first/second-generation TKIs was 73.3%, which was numerically higher than that of third-generation TKIs (53.3%), although the difference was not significant (*p* = 0.26). Additionally, the DCRs of the most recent first-/second-generation and third-generation TKIs were 93.3% and 80%, respectively (*p* = 0.29). Regarding the PFS of the most recent TKI, there were no differences between the first/second-generation group and the third-generation group (7.0 versus 7.2 month; *p* = 0.15). The treatment PFS of the most recent TKI and the best responses are shown in Figure 3.

There was no metastatic tropism of *MET*-amplified cancer during progression after EGFR-TKI therapy. The most common site of progression was primary lung lesions (*n* = 23), followed by intrathoracic lesions, which were indicative of lymph node, pleura, and secondary lung metastases (*n* = 17). Among the biopsy sites from which *MET*-amplified tissue was obtained, lung lesions were the most frequent site for rebiopsy (*n* = 17), followed by the liver because of its easy accessibility. The intrathoracic lymph node, extrathoracic lymph node, and skin were the rest of the sites for rebiopsy, as shown in Appendix A.

## 4. Discussion

Predictive genomic markers, particularly *EGFR* mutant status, are indicative of the efficacy of EGFR-TKI therapy in advanced NSCLC and are now routinely evaluated in clinical practice [20]. Additionally, after acquisition of resistance to first/second-generation EGFR-TKIs, it is now considered routine practice to rebiopsy the progression site for T790M mutation by locked nucleic acid-based assays to determine whether initiation of third-generation EGFR-TKIs is appropriate [21,22]. Because the T790M mutation is the most common mechanism (50–60%) for acquired resistance to first/second-generation TKIs, it is rational to screen for the presence of this mutation routinely [23]. However, other determinants of resistance cannot be as easily screened since they are so rare [24,25].

In our study, the proportion of *MET*-positive patients was 16.1%; in the first/second-generation and third-generation TKI failure groups, 12.1% and 24.2% of patients were positive for *MET*, respectively, consistent with previous studies [13,24,25,26]. Despite the limitations of our study as a single-center study of an unselected *EGFR* mutant lung cancer cohort, this consistent proportion of *MET* amplification as determinant of resistance makes it reasonable to find the enriched population to evaluate. To the best of our knowledge, this is the first *MET* ISH study to confine its scope to patients who developed resistance to prior EGFR-TKI failure, which we commonly encounter in our clinical practice. Although first-line osimertinib treatment is the standard of care in many countries, in Korea, it is not often selected as 1^st^ line TKI because it is not covered by the National Insurance Service of Korea health insurance. Thus, this point should be considered because it is unclear whether the characteristics of the present patients are predictive of *MET* amplification in resistance after first-line therapy in osimertinib.

In this study, *MET* positivity occurred more frequently in ever-smokers who had a history of tobacco use than in never-smokers. Typically, other driver mutations (e.g., *EGFR*, *ALK*, and *ROS*) of lung cancer tend to occur in never-smokers [17,27,28]. Some reports have described a high incidence of ever-smokers in patients with *MET* exon 14 mutation [29,30] and in patients with *MET* amplification [31]. However, another study assumed that patients with *MET* exon 14 mutations whose smoking statuses were not known would be mostly never-smokers owing to their low tumor mutation burden [29]. Although further comprehensive profiling of patients with *MET* amplifications is needed, considering real-world data, it is reasonable to screen patients with a history of tobacco use for *MET* amplification.

Lung cancer easily metastasizes to the brain; indeed, 10–50% of all patients with lung cancer develop brain metastases during their disease course [32,33]. Moreover, patients with *ALK*-positive lung cancer have a higher incidence of brain metastasis, whereas patients with *ROS1*-positive lung cancer have a lower incidence [34,35,36]. To date, no data have described the preferred metastatic site of *MET*-positive lung cancer after previous TKI failure. A recent study in a treatment-naïve population showed that there were no differences in the presence of brain metastasis between *MET*-high and *MET*-low patients, as defined by copy number gain [37]. However, in our study, tropism toward the brain was less prominent in patients with *MET* amplification than in patients without amplification. Our findings suggested a need to identify pathways other than *MET* amplification as resistance mechanisms associated with EGFR-TKI failure in patients having brain metastasis.

Last but not least we found a significant tendency toward a shorter PFS for the most recent TKI in patients with *MET* amplification compared with that in patients without amplification. Some reports have shown that various oncogenic *MET*-driven cancers are associated with poor prognoses [38,39,40]. However, few studies have evaluated PFS associated with previous EGFR-TKIs in patients with *MET* amplification. A recent study of treatment-naïve *EGFR* mutant lung cancer showed that there were no differences in time to treatment failure (TTF) between two groups distinguished according to *MET* copy number. Instead, patients with *MET* amplification showed short TTF and poor outcomes (median TTF: 5 months; range, 1.0–6.4 months) [37]. Although different populations and parameters were used, our results were consistent with the previous results in terms of the shorter PFS in patients with *MET* amplification. Overall, we believe that it is reasonable to perform *MET* ISH in patients who show suboptimal responses to previous EGFR-TKIs in practice.

Clear-cut relationships between *MET* amplification, mutation, and overexpression have not yet been confirmed when collectively applied as predictive markers for *MET*-targeting therapy. Owing to failed results of clinical trials in which patients with *MET*-overexpressing tumors, measured by immunohistochemistry of *MET* protein, were enrolled [41], the preferred biomarker for recent *MET*-TKI clinical trials is gene amplification [15,42]. In addition to codrivers in NSCLC, *MET* mutations are generally thought to be mutually exclusive with mutations in other major lung cancer drivers and have not been shown to be associated with acquired resistance to EGFR-TKI therapy [43]. Because the de novo prevalence of *MET* amplification [44,45] and overlap by verifying degrees with other oncogenic drivers [46] are confounding factors that need to be studied in greater detail, our *MET* ISH results for EGFR-TKI-resistant lung cancer have clinical relevance in routine practice.

This study provided the largest series of *MET* analysis data in patients with lung cancer who developed resistance after EGFR-TKI therapy. However, there are still several limitations to this study. Given the retrospective nature of the study and the heterogeneity of the real world, this study was subject to potential biases. Additionally, because of the low discovery rate of *MET* positivity and the diversity of treatments, it was difficult to perform well-controlled analyses. Therefore, additional studies based on stratification of the generation of EGFT-TKIs and line of therapy should be conducted.

In conclusion, our results revealed the distinct clinical characteristics of patients with *MET* amplification-positive NSCLC after acquisition of resistance to EGFR-TKI therapy. Although *MET* amplified NSCLC is rare, our clinical predictions could aid physicians in identifying patients eligible for *MET* amplification screening and therapeutic targeting. Further efforts are required to standardize the diagnostic method and improve patient access to screening.

## Figures and Tables

**Figure 1 cancers-13-03096-f001:**
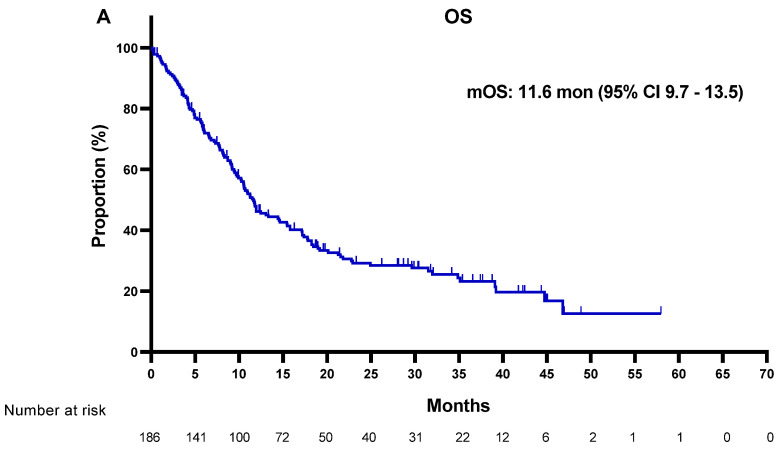
Kaplan–Meier plot for the entire study population. (**A**) Overall survival calculated from the date of *MET* ISH to the last follow-up date. (**B**) Progression-free survival of most recent tyrosine kinase inhibitor used. mOS, median overall survival; mPFS, median progression-free survival; CI, confidence interval; mon, month.

**Figure 2 cancers-13-03096-f002:**
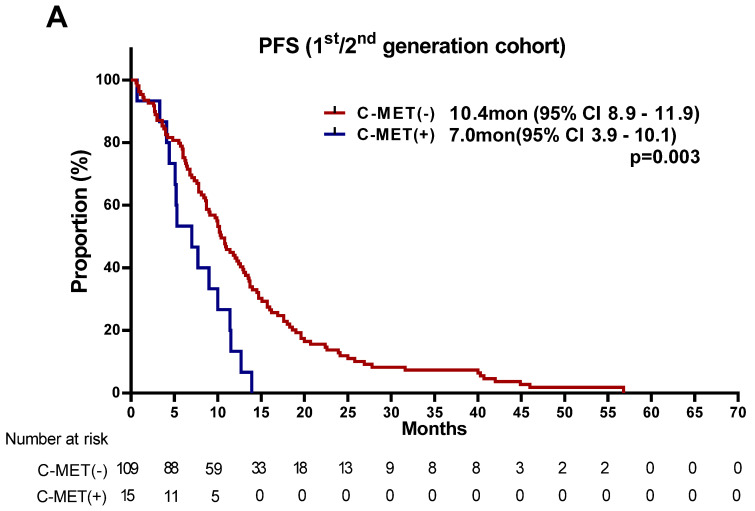
Kaplan-Meier plot of the progression-free survival (PFS) stratified by MET amplification status. (**A**) PFS of 1st/2nd generation tyrosine kinase inhibitor cohort. (**B**) PFS of 3rd generation tyrosine kinase inhibitor cohort. MET, mesenchymal epithelial transition factor; CI, confidence interval; mon, month.

**Figure 3 cancers-13-03096-f003:**
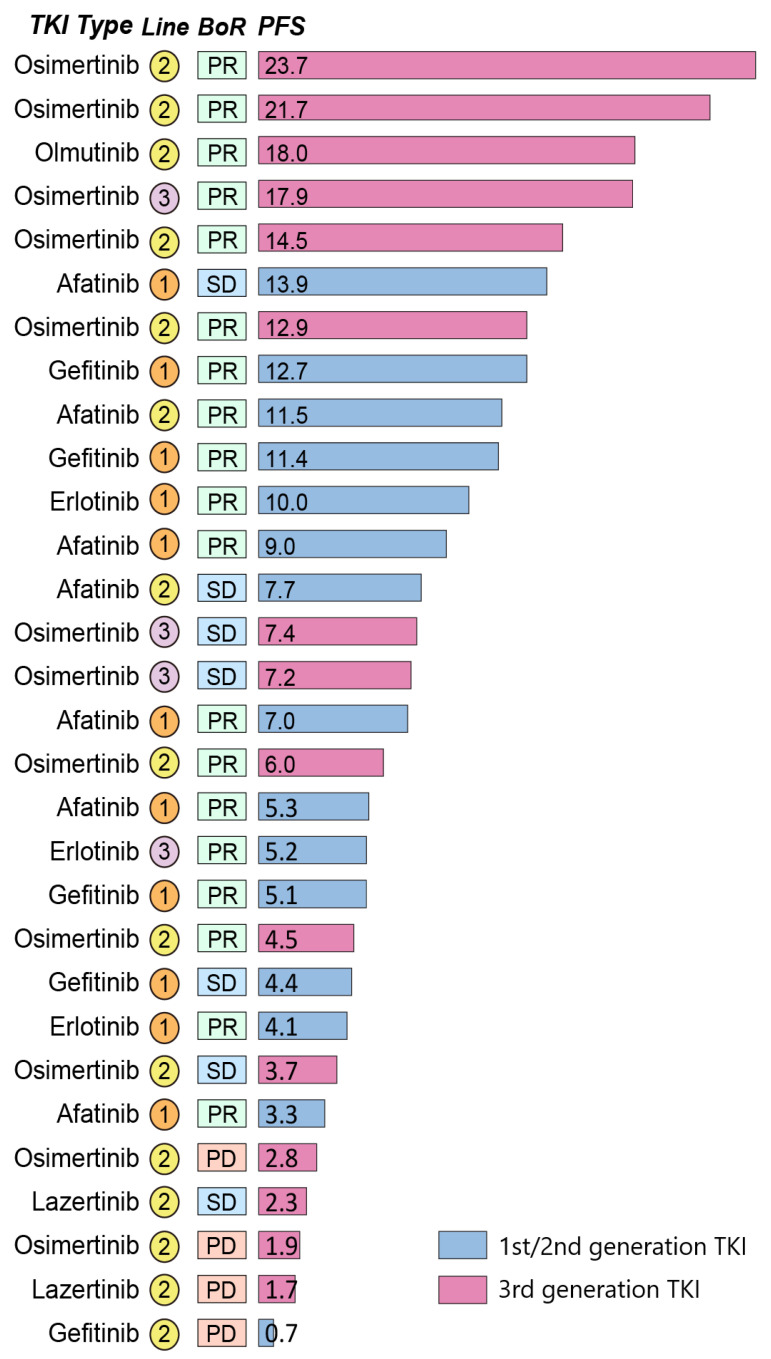
Treatment response with the most recent tyrosine kinase inhibitor. Response indicates the best response to the treatment. The number in the bar indicates progression-free survival in months. TKI, tyrosine kinase inhibitor; BoR, best of response; PFS, progression-free survival; PR, partial response; SD, stable disease; PD, progressive disease.

**Table 1 cancers-13-03096-t001:** Patient characteristics based on *MET* status (*N* = 186).

	MET − (*N* = 156)	MET + (*N* = 30)	
	*N* (%)	*N* (%)	*p* Value
Median age, years (range)	61 (28–84)	59 (28–76)	0.362
Sex			
Male	57 (36.5%)	13 (43.3%)	
Female	99 (63.5%)	17 (56.7%)	0.482
Smoking status			
Current smoker	8 (5.1%)	1 (3.3%)	
Ex-smoker	37 (23.7%)	15 (50.0%)	
Never smoker	111 (71.2%)	14 (46.7%)	0.013
Median pack-year of smoking (interquartile range)	0.0 (0.0–9.5)	2.5 (0–21.0)	0.785
Previous TKI generation			
First/second generation	109 (69.9%)	15 (50%)	
Third generation	47 (30.1%)	15 (50%)	0.034
Previous TKI line			
First line	93 (59.6%)	11 (36.7%)	
Second line	58 (37.2%)	14 (46.7%)	
Third line	5 (3.2%)	5 (16.7%)	0.003
Founder *EGFR* mutation			
Exon 19 deletion	88 (56.4%)	19 (63.3%)	
L858R	59 (37.8%)	10 (33.3%)	
Other mutations **	9 (5.8%)	1 (3.3%)	0.732
Liver metastases			
No metastases	125 (80.1%)	18 (60.0%)	
Baseline metastases without progression	1 (0.6%)	3 (10.0%)	
Baseline metastases with progression	4 (2.6%)	7 (23.3%)	
Progression with new lesion	26 (16.7%)	2 (6.7%)	0.185 *
Brain metastases			
No metastases	44 (28.2%)	16 (53.3%)	
Baseline metastases without progression	7 (4.5%)	7 (23.3%)	
Baseline metastases with progression	59 (37.8%)	5 (16.7%)	
Progression with new lesion	37 (23.7%)	2 (6.7%)	<0.001 *
Not evaluated	9 (5.8%)	0 (0.0%)	

MET, mesenchymal epithelial transition factor; TKI, tyrosine kinase inhibitor; EGFR, epidermal growth factor receptor; * Between progression versus no progression; ** Other mutations include exon 20 insertion (p.A767_V769dup) (*N* = 1), G719X (*N* = 2), L861Q (*N* = 4), and S768I (*N* = 2) in MET (−) group and G179S/L861Q (*N* = 1) in MET (+) group.

**Table 2 cancers-13-03096-t002:** Logistic regression analysis of clinical factors predicting *MET* amplified status.

Category	Univariate			Multivariate		
	HR	95% CI	*p* Value	HR	95% CI	*p* Value
Age	0.983	0.948–1.020	0.360	-	-	-
Sex (male versus female)	0.753	0.341–1.663	0.483	-	-	-
Smoking status (never versus ex- or current smoker)	2.819	1.271–6.252	0.011	3.346	1.326–8.442	0.011
TKI generation (first/second versus third)	2.319	1.049–5.126	0.038	2.618	1.044–6.565	0.040
Baseline *EGFR* mutation site						
Exon 19 deletion	1					
L858R	0.785	0.341–1.807	0.569	-	-	-
Other mutations *	0.515	0.061–4.307	0.540	-	-	-
PFS of most recent TKI	0.930	0.875–0.988	0.019	0.898	0.835–0.965	0.004
Liver metastases (no PD versus PD)	1.800	0.749–4.325	0.189	-	-	-
Brain metastases (no PD versus PD)	0.162	0.065–0.402	<0.001	0.139	0.052–0.373	<0.001

HR, hazard ratio; CI, confidence interval; TKI, tyrosine kinase inhibitor; EGFR, epidermal growth factor receptor; PFS, progression-free survival; PD, progression of disease. * Other mutations; exon 20 insertion (p.A767_V769dup), G719X, G719S/L861Q, L861Q, and S768I.

**Table 3 cancers-13-03096-t003:** Overall response based on latest TKIs of MET-positive patients (*N* = 30).

			Treatment Response			PD Site					
Tyrosine Kinase Inhibitor (TKI)	n	Treatment Line	Best Response ORR DCR	Median PFS(95% CI)	Median OS (95% CI)	Primary Lung	Intrathoracic	Liver	Bone	Extrathoracic	Brain
First-generation TKI	8	1L 6 Pts (75.0%) 2L 1 Pts (12.5%) 3L 1 Pts (12.5%)	PR 6 Pts (75.0%) SD 1 Pts (12.5%) PD 1 Pts (12.5%) ORR 75.0%/DCR 87.5%	5.1 months (4.0–6.1)	81.7 months (NR)	7 (87.5%)	7 (87.5%)	4 (50.0%)	1 (12.5%)	1 (12.5%)	0
Gefitinib	5		ORR 60.0%/DCR 80.0%	5.1 months (3.7–6.4)	NR	4 (80.0%)	5 (100%)	3 (60.0%)	0	1 (20%)	1 (20%)
Erlotinib	3		ORR 100%/DCR 100%	5.2 months (3.5–6.9)	18.7 months (NR)	3 (100%)	2 (66.7%)	1 (33.3%)	1 (33.3%)	0	0
Second-generation TKI Afatinib	7	1L 5 Pts (71.4%) 2L 2 Pts (28.6%)	PR 5 Pts (71.4%) SD 2 Pts (28.6%) ORR 71.4%/DCR 100%	7.7 months (6.0–9.4)	26.5 months (0–71.3)	4 (57.1%)	3 (42.9%)	2 (28.6%)	0	1 (14.3%)	2 (28.6%)
Third-generation TKI	15	2L 12 Pts (80.0%) 3L 3 Pts (20.0%)	PR 8 Pts (53.3%) SD 4 Pts (26.7%) PD 3 Pts (20.0%) ORR 53.3%/DCR 80.0%	7.2 months (3.5–10.8)	38.8 months (31.0–46.6)	12 (80%)	7 (46.7%)	3 (20%)	3 (20%)	6 (40%)	4 (26.7%)
Osimertinib	12		ORR 58.3%/DCR 83.3%	7.2 months (4.8–9.6)	34.7 months (27.6–42.0)	9 (13.3%)	6 (50%)	2 (16.7%)	3 (25%)	5 (41.7%)	3 (25%)
Olmutinib	1		ORR 100%/DCR 100%	18.0 months	74.0 months	1 (100%)	1 (100%)	1 (100%)	0	0	1 (100%)
Lazertinib	2		ORR 0%/DCR 50.0%	1.7 months (NR)	17.0 months (NR)	2 (100%)	0	0	0	1 (50%)	0

ORR, overall response rate; DCR, disease control rate; PFS, progression-free survival; CI, confidence interval; L, line; Pts, patients; PR, partial response; SD, stable disease; PD, progressive disease; NR, not reached.

## Data Availability

The data that support the findings of this study are available from the corresponding author upon reasonable request.

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
