# Peer review of "Distinct Characteristics and Clinical Outcomes to Predict the Emergence of MET Amplification in Patients with Non-Small Cell Lung Cancer Who Developed Resistance after Treatment with Epidermal Growth Factor Receptor Tyrosine Kinase Inhibitors"

_cancers, 2021, doi:10.3390/cancers13123096_

Round 1
Reviewer 1 Report
Ahn et al. are examined the clinical factors that predict the presence of MET amplification after EGFR-TKI resistance. Recently, there are some clinical studies showing that the combination of EGFR-TKI and MET inhibitor is promising in NSCLC harboring EGFR mutation and MET amplification, and it is expected to be used in clinical practice. Therefore, if we can predict MET amplification from a clinical background, it will be good information to consider the necessity of re-biopsy, but this study has many biases and it is difficult to predict based on this result alone.
My comments are listed below.
Major comments:
- Although osimertinib treatment is still regulated in some countries, first-line osimertinib treatment is the standard of care in many countries. As noted by the authors in the text, almost all patients in the third-generation TKI cohort had coexisting EGFR T790M mutations at the time of MET ISH assay, suggesting that most third-generation TKIs are used after second-line therapy. So it is unclear whether the characteristics of the present patients are predictive of MET amplification in resistance after first-line therapy in osimertinib, and it would be better to state this in the discussion.
- After the third-generation EGFR-TKI treatment, MET amplification and EGFR T790M mutation coexist in 15 of the 62 patients which proportion is around 25%. This is relatively low compared with previous reports? (In terms of MET overexpression, Gou et al. reported that the rate of coexistence of EGFR T790M mutation with MET overexpression was 6.8%. Oncotarget. 2016 Aug 9;7(32):51311-51319.)
- Authors consider the lack of progression of brain metastasis at the time of EGFR-TKI resistance to be one of the factors predicting MET amplification, but 15 of the 30 patients(50%) with MET amplification were resistant after the use of third-generation TKIs. In general, Osimertinib has a good central nervous system response, which may be confounded by the progression of brain metastases.
- Is the cut-off you made MET amplification validated?
- In multivariate analysis, TKI generation and TKI line is likely to be cofounded.
- How about describing the Kaplan-Meier plot for OS calculated from the date when MET ISH assay were done or at the progression of EGFR-TKI?
Minor comments:
- In the introduction, the authors mentioned that MET amplification was the most common resistance after the first treatment with osimertinib, but in this study, osimertinib was mostly used after the second line, so the frequency of resistance after the second treatment with osimertinib should be described.
- I think it would be better to describe the median and interquartile range of the number of pack-years of smoke rather than the mean.
- There are a few places where EGFR and MET are not italicized.
Reviewer 2 Report
Summary: MET is a receptor for hepatocyte growth factor in mammalian cells which encodes for a tyrosine kinase, and transduces signaling via MAP kinase and PI-3 kinase pathways. Its aberrant activation contributes to cell proliferation, angiogenesis, and metastasis in many cancers. It is suggested that patients with MET mutations/ amplifications acquire resistance to EGFR inhibitors (TKIs) during treatment in non-small cell lung cancer. In this report, the authors evaluated the occurrence of MET-amplification in patients where failure to treatment with TKIs were observed. They analyzed 186 patients with advanced EGF-R mutant NSCLC for MET amplification status by in situ hybridization (ISH) assay after EGF-R-TKI failure. They performed multivariate logistic regression analysis to evaluate associations between MET amplification status and clinical variables. They report that among EGF-R mutations, 57.7% patients had exon 19 deletion, and 39.2% had exon 21 mutations. The proportions of MET ISH assays performed after first/second-generation and third-generation TKI failure were 66.7% AND 33.1%. The median progression-free survival (PFS) for the most recent EGF-R-TKI treatment was shorter in MET amplification-positive patients than in MET-amplification-negative patients. Their analysis also suggested that smoking history, short PFS on the most recent TKI, and less intracranial progression were associated with a high probability of MET amplification.
Major points:
- On page 4, line 51-52, the authors state that Patient characteristics based on MET amplification status are shown in Table 1. I do not see any table in the paper.
- On page 4, line 161, the authors direct the readers to Fig. 1A. I do not see Fig. 1A embedded in the text.
- On page 4, line 170, the authors direct the readers to Fig. 1B. I do not see Fig. 1B embedded in the text.
- On page 4, line 192, the authors direct the readers to Table 3. I do not see Table 3 embedded in the text.
- In the text, no figures and tables to support the findings are given.
Author Response
RESPONSE TO COMMENTS FROM REVIEWER #2
COMMENTS
- On page 4, line 51-52, the authors state that Patient characteristics based on MET amplification status are shown in Table 1. I do not see any table in the paper.
- On page 4, line 161, the authors direct the readers to Fig. 1A. I do not see Fig. 1A embedded in the text.
3.On page 4, line 170, the authors direct the readers to Fig. 1B. I do not see Fig. 1B embedded in the text.
4.On page 4, line 192, the authors direct the readers to Table 3. I do not see Table 3 embedded in the text.
5.In the text, no figures and tables to support the findings are given.
RESPONSE
We heard that Reviewer #2 reviewed the previous version manuscript of which the tables and figures were in the separate file. After 1st revision, revised paper will be sent to Reviewer #2 again.
Round 2
Reviewer 2 Report
MET is a receptor for hepatocyte growth factor in mammalian cells which encodes for a tyrosine kinase, and transduces signaling via MAP kinase and PI-3 kinase pathways. Its aberrant activation contributes to cell proliferation, angiogenesis, and metastasis in many cancers. It is suggested that patients with MET mutations/ amplifications acquire resistance to EGFR inhibitors (TKIs) during treatment in non-small cell lung cancer. In this report, the authors evaluated the occurrence of MET-amplification in patients where failure to treatment with TKIs were observed. They analyzed 186 patients with advanced EGF-R mutant NSCLC for MET amplification status by in situ hybridization (ISH) assay after EGF-R-TKI failure. They performed multivariate logistic regression analysis to evaluate associations between MET amplification status and clinical variables. They report that among EGF-R mutations, 57.7% patients had exon 19 deletion, and 39.2% had exon 21 mutations. The proportions of MET ISH assays performed after first/second-generation and third-generation TKI failure were 66.7% AND 33.1%. The median progression-free survival (PFS) for the most recent EGF-R-TKI treatment was shorter in MET amplification-positive patients than in MET-amplification-negative patients. Their analysis also suggested that smoking history, short PFS on the most recent TKI, and less intracranial progression were associated with a high probability of MET amplification.
Author Response
Thank you for your insightful comments and review. We revised the manuscript as reviewers and editors indicated.